

# Estimation of water yield in the hydrographic basins of southern Ecuador

Saula Minga-León [1], Miguel Angel Gómez-Albores [1], Khalidou M. Bâ [1], Luis Balcázar [1], Luis Ricardo Manzano-Solís [2], Angela P. Cuervo-Robayo [3,4], Carlos Alberto Mastachi-Loza [1]

[1]Instituto Interamericano de Tecnología y Ciencias del Agua, Universidad Autónoma del Estado de México, Toluca 50200, México
[2]Facultad de Geografía, Universidad Autónoma del Estado de México, Toluca 50110, México
[3]Comisión Nacional para el Conocimiento y Uso de la Biodiversidad (Conabio), Ciudad de México 14010, México
[4]Departamento de Zoología, Instituto de Biología, Universidad Nacional Autónoma de México, Ciudad de México 04510, México

*Correspondence to*: Miguel Angel Gómez-Albores (magomeza@uaemex.mx)

**Abstract.** Humans greatly benefit from natural water resources, also known as hydrological ecosystem services. However, these services may be reduced by population growth, land use changes, and climate change. As these problems become more critical, the need to quantify water resources increases. The estimation of water yield and its distribution are of great importance for the management of water resources. In the present study, the average annual water yield of the hydrographic basins in the southern region of Ecuador was estimated for the 1970–2015 period using the InVEST water yield model based on the Budyko framework. The model estimates annual surface run-off at the pixel, sub-basin, and basin level considering the following variables: precipitation, actual evapotranspiration, land cover/use, soil depth, and available water content for plants. The model was calibrated by varying the ecohydrological parameter $Z$ to reduce error between estimated and observed water yield. The results showed that the modeling of water yield in the majority of the hydrographic basins was satisfactory, allowing the basins to be ranked according to their importance for water production. The Mayo and Zamora basins had the highest water production, corresponding with 934 and 1218 mm per year, respectively, while the Alamor and Catamayo basins had the lowest water production, corresponding with 206 and 291 mm per year, respectively. The present study provides an initial estimate of water yield at the basin level in the southern region of Ecuador, and the results can be used to evaluate the impacts of land cover changes and climate change over time.

## 1 Introduction

Humans receive a variety of benefits from natural water resources, also known as hydrological ecosystem services (HESs). Several of the main HESs include regulation of the water cycle, high water yields, maintenance of water quality, and aquifer recharge. These services can be negatively affected if natural ecosystems are converted to other land uses by human activities (CONDESAN and IEP, 2010). Decades of anthropogenic intervention have altered land covers and uses, resulting in the conversion of natural lands to pastures, crops, deforested areas, urban constructions, and mining operations, among



other land uses (Crespo et al., 2014). As these problems become more critical, effective land use planning is necessary to guarantee the continued provision of HESs. Such planning requires a good understanding of hydrology and can also be used to support key decisions about the management of water resources in the near future (Buytaert et al., 2006b). The uncertainty associated with climate change and its impacts on the water cycle and water availability further incentivize the need to

generate detailed inventories of current water resources (Tallis and Polasky, 2009).

In Ecuador, few studies have quantified water resources at the basin level (Buytaert et al., 2006b; Duque Yaguache and Vázquez Zambrano, 2015). One of the main obstacles in advancing hydrological knowledge is the difficulty of implementing and maintaining observation networks in complex or remote environments as well as the lack of recognition of ecosystems as providers of water (Célleri et al., 2009). Some microbasins in the wet paramos and tropical mountains of southern

Ecuador have been studied by researchers (e.g., Buytaert et al., 2005, 2006a; Célleri et al., 2009; Crespo et al., 2014, 2011; Fleischbein et al., 2006; Vázquez, 2015), whose main contributions have been toward the conceptualization of hydrological processes, the monitoring and quantification of several important climate variables, and the analysis of hydrophysical soil properties. However, these studies have generally been carried out at small scales (Duque Yaguache and Vázquez Zambrano, 2015) and over relatively short periods of time. Their focus has mostly centered on the importance of Andean ecosystems as

the main suppliers of water for the Andean highlands as well as for the extensive neighboring lowland areas and coastal plains (Buytaert et al., 2006b). Crespo et al. (2014) highlighted that water production in the microbasins of natural paramos is around 75% of the precipitation level. This high water production is due to the high level of precipitation and the low consumption of water by vegetation. However, urban growth and the concentration of agriculture in the inter-Andean valleys in addition to other land use changes have directly contributed toward an increase in water demand in the region.

The modeling of hydrological services is associated with a considerable implementation effort and data requirement, and the required data may not always be available. For this reason, it is necessary to explore accessible tools that require little information but can still provide a good overall scenario of available hydrological services. In particular, to quantify available water resources, traditional hydrological and ecosystem service tools are available. The InVEST model is one tool for the valuation of ecosystem services developed in 2007 by Stanford University, the World Wildlife Fund (WWF), and the

Nature Conservancy (TNC). It contains several sub-models, including a model for estimating water yield. This latter model has a biophysical component that calculates water yield based on landscape features in addition to a valuation component that aims to represent the benefits of the available water supply for humans (Hamel and Guswa, 2015).

The biophysical module used in this work is based on the Budyko hydrological framework. This model estimates annual average run-off at the pixel, sub-basin, and basin level, considering variables such as precipitation, reference

evapotranspiration, land use and cover, soil depth, and available water content for plants. The InVEST water yield model has been widely used throughout the world with satisfactory results in China (e.g., Bai et al., 2012; Geng et al., 2015; Lang et al., 2017; Li et al., 2018; Zhang et al., 2012), Thailand ( e.g., Arunyawat and Shrestha, 2016), Spain ( e.g., Bangash et al., 2013), Argentina ( e.g., Gaspari et al., 2015; Izquierdo and Clark, 2012), and several countries in northeastern Africa ( e.g., Belete et al., 2018). Other studies such as those of Sánchez-Canales et al. (2012), Hamel and Guswa (2015), and Redhead et al.



(2016) have analyzed the sensitivity, calibration, and validation of the model and have agreed that it produces accurate estimates of water production in hydrographic basins. However, the uncertainty introduced by errors in climate data can be significant and non-spatially heterogeneous, subsequently affecting the estimated spatial distribution of water yield.

Given this context, the objective of the present study was to estimate and map average annual water yield for the 1970–2015

period in nine hydrographic basins of the southern region of Ecuador. The largest amount of meteorological, precipitation, and temperature data available for the region was used. Given the interest in water yield from the perspective of ecosystem services, the InVEST water yield model was used to generate explicit spatial estimates of water yield. The results are an initial estimate of water yield at the hydrographic basin level and can be used to evaluate the impact of land cover changes and climate change and to thereby facilitate the planning and local management of water resources in the long term.

## 2 Materials and methods

### 2.1 Study area

The study was carried out in nine hydrographic basins of the southern region of Ecuador located in the provinces of El Oro, Loja, and Zamora Chinchipe that, as a whole, form Planning Zone 7 (Zona de Planificación Siete [ZP7]), a political and administrative unit (Figure 1). This unit has an area of ~27491 km$^2$ and is located between 3°02' and 5°00' N and 78°21' and

80°29' W. The altitude ranges from 0 to ~3500 masl.

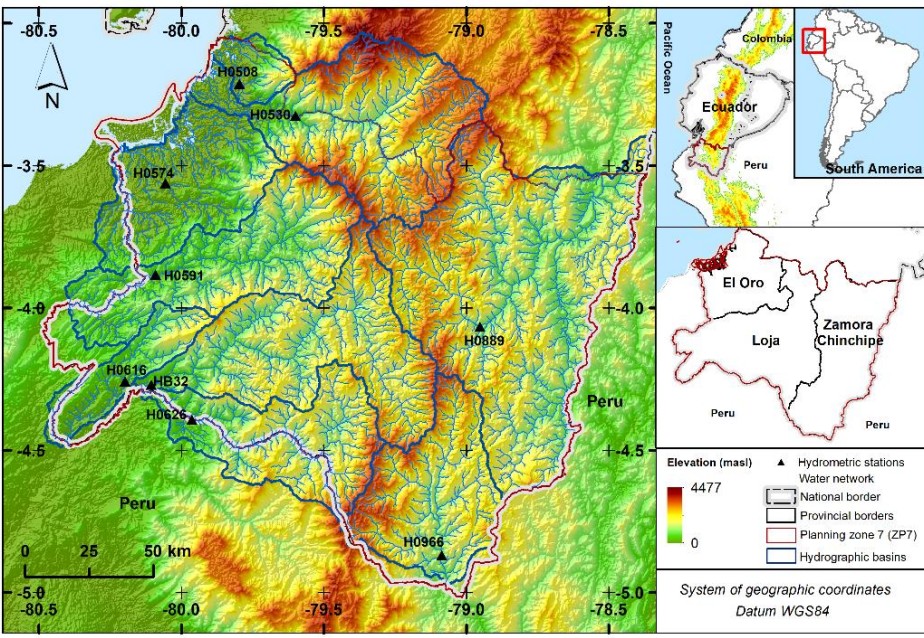

**Figure 1: Location of the study area and the hydrometric stations.**


Sixteen climate types are present in the study area according to the Thornthwaite climate classification performed by the Instituto Nacional de Meteorología e Hidrología (INAMHI) (Moya, 2006). Some of the main factors that influence the climate are oceanographic phenomena, such as the El Niño–Southern Oscillation and the Humboldt current (Maldonado A., 2001), and the presence of the Andes mountain range. Specifically, the study area coincides with one the lowest sectors of

the mountain range known as the inter-Andean depression (Figure 1). In addition, the region is considered as the limit between the northern and central Andes. The high sectors are extremely wet throughout the year, whereas the downwind slopes, in contrast, are relatively dry. The region is also characterized by an extreme vertical precipitation gradient and the fragmented relief of valleys and mountain ranges, which further influence the significant climate heterogeneity of the region (Richter and Moreira, 2005).

Precipitation (Figure 2a) in the hydrographic basins on the Pacific side of the Andes ranges from ~500 to 1500 mm/year and on the Amazon side of Andes ranges from ~1000 to 2400 mm/year. Temperature (Figure 2b) in both areas varies from 6.5 to 25.5 °C. Areas closer to the Andes mountain range are colder, and those closer to the coast are warmer.

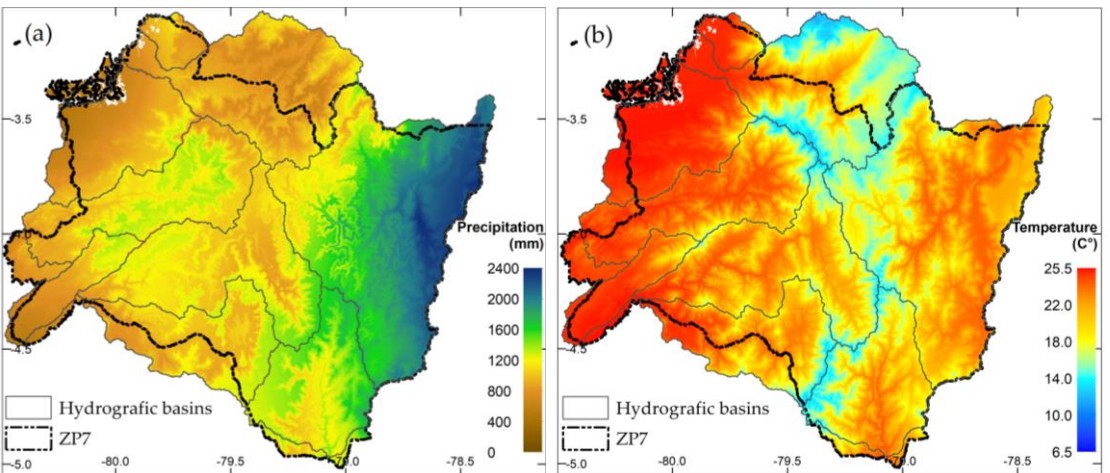

**Figure 2: Spatial distribution of (a) average annual precipitation and (b) average annual temperature in the study area for the**
**1970–2015 period.**

**2.2 InVEST water yield model**

The InVEST water yield (WY) model is based on the Budyko curve and on average annual precipitation (Eq. 1). The model calculates average surface run-off (Y) of pixel $x$ in the landscape as follows:

$$Y_x = \left(1 - \frac{\text{AET}_x}{P_x}\right) \times P_x \qquad\qquad (1)$$

where $\text{AET}_x$ is the annual actual evapotranspiration of pixel x, and $P_x$ is annual precipitation of pixel x. The fraction $\text{AET}_{xj}/P_x$ is based on the expression of the Budyko curve proposed by Fu (1981, cited in Sharp et al. 2018) and Zhang et al. (2004) (Eq. 2):



$$\frac{AET_x}{P_x} = 1 + \frac{PET_x}{P_x} - \left[1 + \left(\frac{PET_x}{P_x}\right)^{\omega_x}\right]^{1/\omega_x} \qquad (2)$$

where $PET_x$ is the potential evapotranspiration, and $\omega_x$ is a non-physical parameter that characterizes the natural climate-soil properties proposed by Donohue et al. (2012). $PET_x$ (Eq. 3) and $\omega_x$ (Eq. 4) are defined as follows:

$$PET_x = K_{c\,(l_x)} \times ET_{0\,x} \qquad (3)$$

$$\omega_x = Z \times \frac{AWC_x}{P_x} + 1.25 \qquad (4)$$

where $ET_{0\,x}$ is the reference evapotranspiration of pixel $x$, and $K_c$ is the evapotranspiration coefficient of vegetation associated with a specific land use/cover (LUC) $(l_x)$. $ET_0$ reflects local climate conditions as a function of the evapotranspiration of a reference vegetation. $K_c$ adjusts the $ET_0$ values according to the vegetation type of pixel x.

$AWC_x$ is the available water content for plants in mm and is defined based on soil texture and effective root depth. $AWC_x$

establishes the quantity of water that can be stored and liberated in the soil for plant use. It is estimated as the product of the fraction of plant available water capacity (PAWC) and of root restricting layer depth and vegetation rooting depth (Eq. 5):

$$AWC_x = \text{Min}\,(\text{Rest.layer.depth, root.depth}) \times PAWC \qquad (5)$$

The depth of the root restriction layer is the soil depth at which the penetration of roots is inhibited because of physical or chemical characteristics. The depth of the rooting zone is often determined as the depth above which 95% of root biomass of

a particular vegetation type is located. The PAWC is the available water capacity for plants, which is to say, the difference between the field capacity and the wilting point.

The $Z$ value is an empirical constant that captures basin characteristics such as climate, precipitation intensity, and topography. The minimum value of $\omega_x$ is 1.25, which is the value for bare soil (when root depth is 0). Three methods were proposed by Sharp et al. (2018) for the determination of Z. The first method is based on the positive correlation between $Z$

and average annual precipitation events per year (N), as $Z$ approximates to N/5 (Donohue et al., 2012). This implies that $Z$ captures precipitation patterns and distinguishes between basins with similar annual precipitation but distinct precipitation intensity (Hamel and Guswa, 2015). The second method is based on solving $Z$ in Eq. 4; the necessary ω values are available at the worldwide level (Liang and Liu, 2014; Xu et al., 2013). The third method, which was used in the present study, determines $Z$ through calibration, which consists of comparing estimated data with observed data.

Finally, for LUCs without vegetation (water bodies, urban areas, etc.), AET is directly calculated from $ET_0$ values, and the upper limit is determined by precipitation, as follows (Eq. 6)

$$AET_x = \text{Min}\,(K_c(l_x) \times ET_{0\,x}, P_x) \qquad (6)$$

Accordingly, the WY model calculates average surface run-off per pixel in addition to average surface run-off at the hydrographic sub-basin and basin level. A detailed description of the model can be consulted in Sharp et al. (2018) and

Hamel and Guswa (2015).




## 2.3 Data

The data required by the InVEST WY model are average annual precipitation, annual average reference evapotranspiration ($ETo$), land use/cover (LUC), fraction of plant available water capacity (PAWC), soil depth, and the boundaries of the hydrographic basins (Figure 3) in addition to the biophysical attribute table and the ecohydrological parameter $Z$.

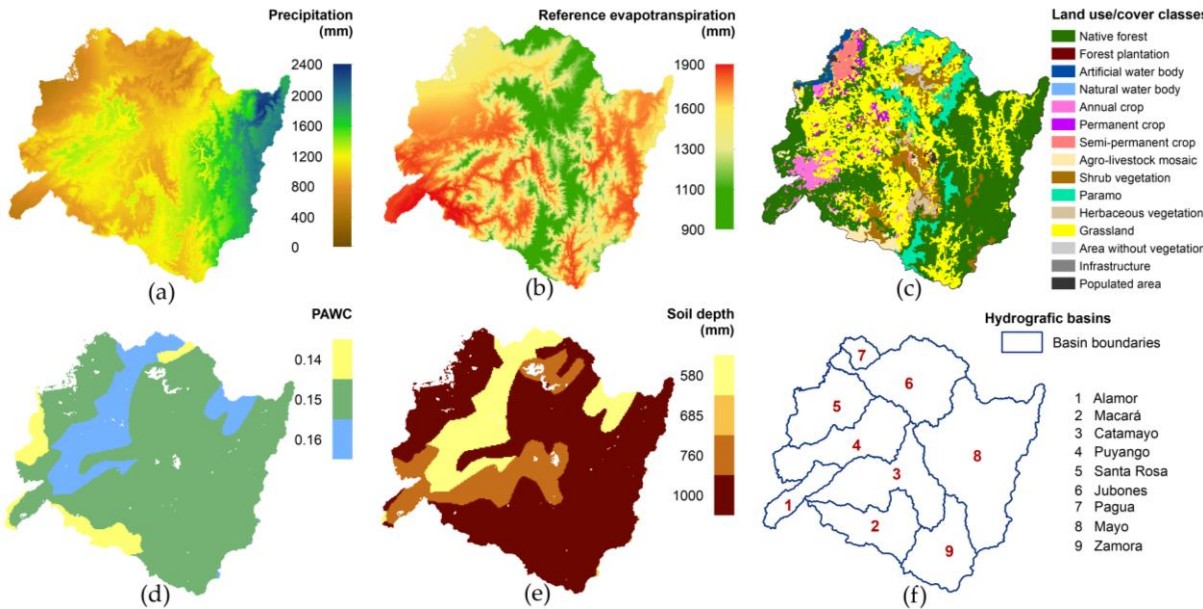

**Figure 3: Variables required for the execution of the InVEST water yield model: (a) precipitation (mm), (b) reference evapotranspiration (mm), (c) land use and cover, (d) plant available water capacity, (d) soil depth, and (f) hydrographic basins.**

**Climate data**: The climate data are the most important input variables in the water yield model. Images of precipitation and of average, minimum, and maximum temperature at a high resolution of 6 arc seconds (~185 m) were generated through interpolation based on observed data using the thin-plate smoothing method in the ANUSPLIN software (Hutchinson, 2006).

The climate data were obtained from the network of meteorological stations of the INAMHI, the Servicio Nacional de Meteorología e Hidrología (SENAMHI) of Peru, and the Estación Científica San Francisco (ECSF). Precipitation data from a total of 160 meteorological stations were used in addition to temperature data from 64 stations. The data were recorded in different formats and at distinct temporal scales, so the data were converted to monthly values for the 1970−2015 period. The data were subsequently structured and validated; this process consisted in (1) merging the data according to the

recommendations of the World Meteorological Organization (WMO, 2011), (2) detecting gaps in the data using graphs, and (3) identifying atypical values. To identify atypical data in the variables of average, maximum, and minimum temperature, an upper limit of 1.5 times the interquartile range (IQR) was established; meanwhile, for precipitation, considering that the variability in the data is much higher, the limit was set at 3 times the IQR.

The temperature images were used to estimate $ET_0$ using the Hargreaves equation (Eq. 7).

$$ET_0 = 0.0023 \ Ra \ \left[\left(\frac{T_{max}+T_{min}}{2}\right)+17.8\right] \ (T_{max}-T_{min})^{0.5} \ \left(\frac{mm}{day}\right) \tag{7}$$




where Ra is terrestrial solar radiation in mm/day, $T_{max}$ is the maximum temperature (°C), and $T_{min}$ is the minimum temperature (°C). Ra was obtained from the databases of Global-Aridity and Global-PET from the Consultative Group on International Agricultural Research-Consortium for Spatial Information (CGIAR-CSI) at a resolution of 1.5 arc minutes. The resolution of the Ra image was adjusted to 6 arc seconds. In the study area, the maximum spatial variation of Ra is 0.35 mm

(13.22–12.87 mm) in June, and the minimum variation is 0.002 mm in March (15.561–15.563 mm), indicating that the variation of Ra across the study area is relatively small and that the resolution adjustment likely did not affect the estimation of $ET_0$.

**Land use and cover (LUC)**: The database on LUC was obtained from the Ministerio del Ambiente del Ecuador (MAE) at a scale of 1:100000 for 2014. From these data, a raster image was generated based on the level-II classification proposed by

MAE and Ministerio de Agricultura, Ganadería, Acuacultura and Pesca (MAGAP) (Ministerio del Ambiente (MAE) and Ministerio de Agricultura, Ganadería, 2015) which contained 15 classes of LUC for the study area (Figure 3c).

**Depth of the root restriction layer**: Data on the root restriction layer were unavailable. For this reason, according to the recommendation of Sharp et al. (Sharp et al., 2018), soil depth was used. Soil depth data were obtained from the Harmonized World Soil Database (HWSD) version 1.2 (FAO et al., 2012). From this, a raster image was generated. The unit of

measurement was millimeters (mm).

**Plant available water capacity (PAWC)**: From the HWSD, a raster image of PAWC (mm) was also generated. The raster data for PAWC were then divided by the raster data for soil depth (mm) to obtain the fraction required for the WY model. The resulting values are dimensionless (0 to 1).

**Biophysical attribute table**: A table containing biophysical attributes for each LUC class (Table 1) was generated in text

format (.csv). The attributes are as follows: Id, LUC class, evapotranspiration coefficient ($K_c$), root depth, and vegetation presence. The values for $K_c$ and root depth were taken from Allen et al. (Allen et al., 1998) and Sharp et al. (Sharp et al., 2018). $K_c$ is used to obtain potential evapotranspiration from $ET_0$; the value ranges from 0 to 1.5. The field vegetation presence determines which equation will be used to estimate actual evapotranspiration ($AET$) in the WY model: A value of 1 corresponds with LUCs with vegetation (Eq. 2), and a value of 0 corresponds with LUCs without vegetation (Eq. 6).

**Ecohydrological parameter Z**: $Z$ is an empirical constant with a value of 1 to 30 that was determined during the calibration, which involved the comparison of estimated data with observed data. At following, the calibration process is described in greater detail.



**Table 1: Biophysical attribute table used to execute the water yield model.**

| Id | LUC classes | $K_c$ | Root depth (mm) | Presence vegetation |
|----|-------------|-------|-----------------|---------------------|
| 1 | Native forest | 1 | 5000 | 1 |
| 2 | Natural water body | 1 | - | 0 |
| 3 | Semi-permanent crop | 0.6 | 1000 | 1 |
| 4 | Grassland | 0.7 | 1000 | 1 |
| 5 | Permanent crop | 0.6 | 1000 | 1 |
| 6 | Shrub vegetation | 0.5 | 2000 | 1 |
| 7 | Agro-livestock mosaic | 0.6 | 1000 | 1 |
| 8 | Populated area | 0.3 | - | 0 |
| 9 | Area without vegetation | 0.3 | - | 0 |
| 10 | Artificial water body | 1 | - | 0 |
| 11 | Paramo | 0.7 | 1500 | 1 |
| 12 | Herbaceous vegetation | 0.7 | 1500 | 1 |
| 13 | Forest plantation | 1 | 5000 | 1 |
| 14 | Annual crop | 0.6 | 900 | 1 |
| 15 | Infrastructure | 0.3 | - | 0 |

## 2.4 Calibration of the Water Yield model

To calibrate the WY model, as stated above, the ecohydrological parameter *Z* and observed data (long-term average water flow) collected from nine hydrological stations were used. Unfortunately, hydrological stations were not present at the outlets of the hydrographic basins; data from these stations are ideal for model calibration. The nine available hydrometric stations were located in sub-basins of the hydrographic basins of the study area (Figure 1)

The calibration was performed up to the elevation of the stations. The *Z* value was varied from 1 to 30 until the error was minimized; in this case, the error was defined as the difference between the estimated and observed water yield. Observed water yield was determined from data on daily water flow from INAMHI. The calibration period of each hydrographic sub-basin depended on the availability of observed data from hydrometric stations (Table 2).





**Table 2: Information on hydrometric stations and calibration period for the water yield model.**

| Station code | Hydrometric station | Hydrographic basin | Calibration period | Evaluated years | Excluded years |
|---|---|---|---|---|---|
| H0616 | Alamor en Saucillo (Dj Celica) | Alamor | 1970–2011 | 42 | - |
| H0626 | Macará en Pte. Internacional | Macará | 1979–1994 | 16 | - |
| HB32 | Catamayo en Vicin | Catamayo | 2006–2014 | 8 | 2012 |
| H0591 | Puyango en Cpto. Militar (Pte. Carretera) | Puyango | 1970–2011 | 42 | - |
| H0574 | Arenillas en Arenillas | Santa Rosa | 1970–2011 | 34 | 1984-1990 |
| H0530 | Jubones en Ushcurrumi | Jubones | 1970–2011 | 42 | - |
| H0508 | Chaguana en Pte. Carretera | Pagua | 2003–2011 | 9 | - |
| H0889 | Zamora DJ Sabanilla (en Zamora) | Zamora | 1982–2011 | 28 | 1985, 2002 |
| H0966 | Mayo AJ Qda. Zumbayacu | Mayo | 1982–2011 | 15 | 1984, 1986-1988, 1993-1995, 1997-2002, 2005-2006 |

The units for the observed data were cubic meters per second ($m^3$/s), and the units of the estimated data were millimeters per year (mm/year). Accordingly, the observed data were transformed to mm/year.

5  In the hydrographic sub-basins, the diversion of water flow by irrigation systems was additionally considered. Only water flows that could influence the calibration of the model were taken into account (Table 3), such as the Zapotillo irrigation system located before station HB32 that diverts approximately 8 $m^3$/s. Small water channels that communities use for irrigation were not considered because information was lacking on the diverted water volume and because the diverted quantities were considered minimal.

10  **Table 3: Irrigation systems considered in the calibration of the water yield model.**

| Hydrographic basin | Station code | Irrigation channels | Total diverted volume ($m^3$/s) | Source | Date |
|---|---|---|---|---|---|
| Macará | H0626 | 7 | 3.29 | PDOT Prov. Loja (Gobierno Provincial de Loja (GADP-Loja), 2013) | up to 2012 |
| Catamayo | HB32 | 7 | 9.40 | PDOT Prov. Loja (Gobierno Provincial de Loja (GADP-Loja), 2013) | up to 2012 |

The calibrated $Z$ values were used to obtain the water yield of each hydrographic basin under the assumption that the calibrated value captured the climate characteristics, precipitation intensity, and topography of the basins.





## 3 Results

### 3.1 Calibration of the water yield model

The model was calibrated for five hydrographic sub-basins with $Z$ values ≥ 3 and errors of less than 7%. Meanwhile, the remaining four sub-basins could not be satisfactorily calibrated, as water production was underestimated in these sub-basins

5  by 20% to 50% (Table 4).

Table 4: Summary of the calibration process per hydrographic sub-basin.

| Station code | Hydrometric station | Area (Km²) | P (mm) | $ET_0$ (mm) | PET (mm) | AET (mm) | Z | Obs. WY (mm) | Est. WY (mm) | Relative error (%) |
|---|---|---|---|---|---|---|---|---|---|---|
| H0616 | Alamor en Saucillo (Dj Celica) | 585 | 1108 | 1660 | 1157 | 725 | 13 | 381 | 383 | 0.58 |
| H0626 | Macará en Pte. Internacional | 2509 | 986 | 1544 | 1124 | 595 | 5 | 360 | 349 | −3.21 |
| HB32 | Catamayo en Vicin | 4172 | 1251 | 1536 | 1170 | 833 | 24 | 303 | 303 | 0.1 |
| H0591 | Puyango en Cpto. Militar (Pte. Carretera) | 2728 | 1149 | 1585 | 1153 | 375 | 1 | 1008 | 775 | −23.18 |
| H0574 | Arenillas en Arenillas | 493 | 1009 | 1678 | 1302 | 533 | 3 | 447 | 477 | 6.61 |
| H0530 | Jubones en Ushcurrumi | 3636 | 870 | 1249 | 810 | 447 | 4 | 423 | 422 | −0.10 |
| H0508 | Chaguana en Pte. Carretera | 190 | 789 | 1505 | 1169 | 352 | 1 | 576 | 437 | −24.19 |
| H0889 | Zamora DJ Sabanilla (en Zamora) | 1422 | 1343 | 1240 | 1070 | 399 | 1 | 1724 | 994 | −45.23 |
| H0966 | Mayo AJ Qda. Zumbayacu | 2564 | 1424 | 1363 | 1189 | 422 | 1 | 1987 | 1001 | −49.60 |

$ET_0$: reference evapotranspiration, *PET*: potential evapotranspiration, *AET*: actual evapotranspiration, Obs. WY: observed water yield, Est. WY: estimated water yield, P: precipitation

### 3.2 Water yield

10  The water yield varies from 206 mm in Alamor basin to 1218 in Zamora basin. The Amazon basins (Zamora and Mayo) have the highest water production, and the basins located in the Andean mountains and valleys (Alamor and Catamayo) have the lowest water production.





**Table 5: Average annual water yield per hydrological basin (1970–2015).**

| Id | Hydrographic basin | Area (Km$^2$) | P (mm) | ET$_0$ (mm) | PET (mm) | AET (mm) | Est. WY (mm) | Volume (Mm$^3$) |
|----|----|----|----|----|----|----|----|----|
| 1 | Alamor | 1187 | 899 | 1752 | 1436 | 693 | 206 | 245 |
| 2 | Macará | 2824 | 1039 | 1577 | 1193 | 607 | 432 | 1220 |
| 3 | Catamayo | 4187 | 1121 | 1528 | 1167 | 863 | 258 | 1080 |
| 4 | Puyango | 3601 | 1106 | 1629 | 1240 | 382 | 724 | 2606 |
| 5 | Santa Rosa | 2708 | 842 | 1603 | 1263 | 505 | 336 | 910 |
| 6 | Jubones | 4402 | 875 | 1284 | 861 | 460 | 415 | 1827 |
| 7 | Pagua | 505 | 906 | 1457 | 1075 | 352 | 553 | 279 |
| 8 | Zamora | 8297 | 1701 | 1518 | 1376 | 482 | 1218 | 10109 |
| 9 | Mayo | 2869 | 1356 | 1390 | 1211 | 421 | 934 | 2681 |

ETo: reference evapotranspiration, PET: potential evapotranspiration, AET: actual evapotranspiration, Est. WY: estimated water yield, P: precipitation

Precipitation and $ET_0$ were similar in some basins, for example, Macará and Catamayo; however, the A$ET$ differed, generating distinct water yields (Table 5). This can mainly be attributed to the influence of $K_c$ in the estimation of $AET$. The value of $K_c$ depends on the type of vegetation cover. Lower values of $K_c$ correspond with lower values of A$ET$ and, consequently, with higher values of water production (and vice versa). The model assumes that all water in excess of water lost due to $AET$ is water production.

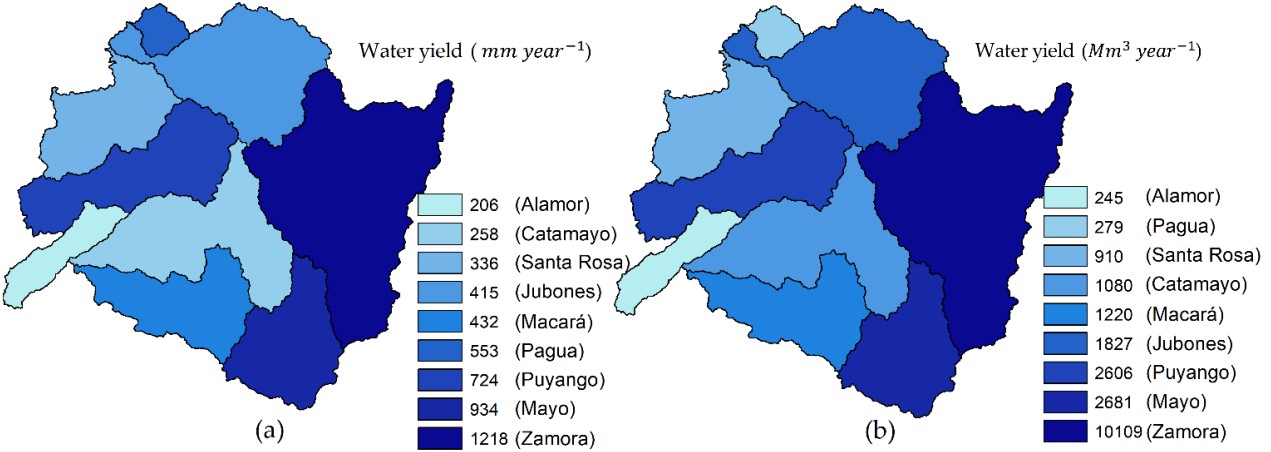

**Figure 4: Spatial distribution of average annual water yield in the hydrographic basins of the study area from 1970–2015: (a) surface run-off depth (mm) and (b) water volume (mm$^3$).**

Water production at the pixel level is shown in

Figure 5, which illustrates the zones of high and low water production per hydrographic unit. Comparing water yield with the spatial distribution of precipitation (Figure 2), the water yield generally increases in areas of high precipitation. For example, the highest precipitation occurs in the eastern portion of the Zamora basin, where the water production also tends to be higher.





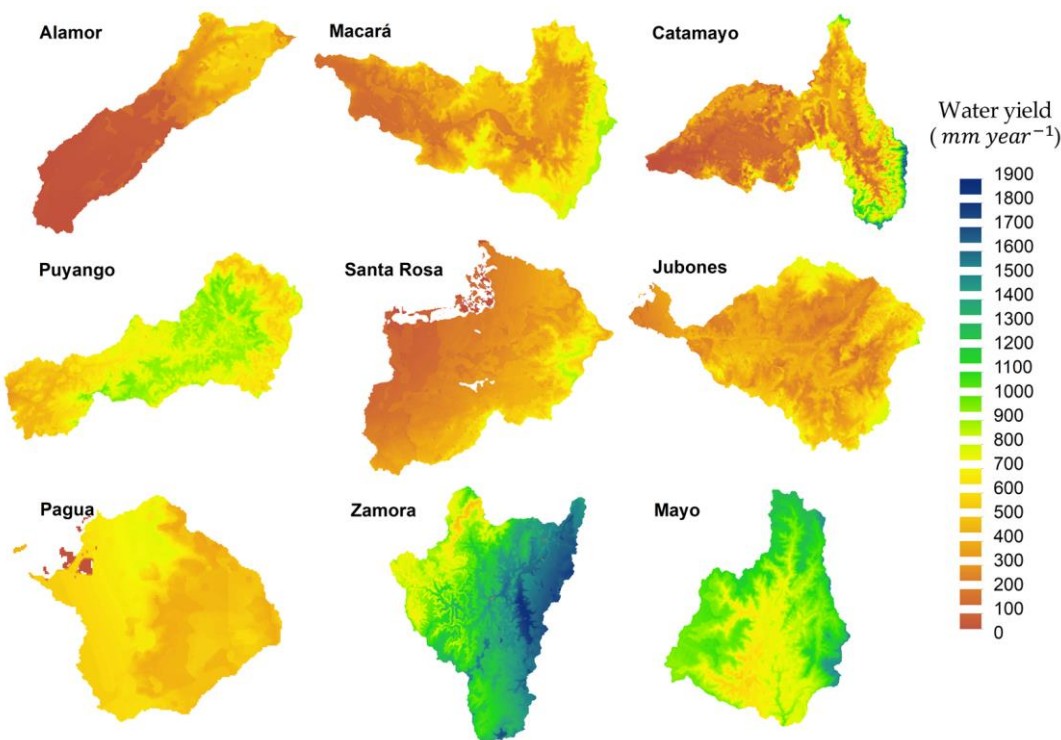

Figure 5: Spatial distribution of average annual water yield in the hydrographic basins of the study area from 1970–2015.

## 4 Discussion

### Calibration of the Water Yield model

The InVEST water yield model was calibrated for the five hydrographic sub-basins with errors of less than 7%. Four sub-basins were unable to be calibrated because the results for water yield were underestimated. This result may be attributed to the precipitation data, as precipitation is the most sensitive variable in the estimation of water yield according to several studies (Goyal and Khan, 2016; Hamel and Guswa, 2015; Redhead et al., 2016; Sánchez-Canales et al., 2012). Notably, Rollenbeck and Bendix (2011) evaluated a set of local area weather radar (LAWR) images in southern Ecuador and found

that precipitation reached 4000 mm in the high mountains. However, the images obtained in the present study from the interpolation indicated a precipitation level of 2400 mm for this same area.

As previously mentioned, the underestimation of precipitation is possibly due to the low density of meteorological stations in the high mountain zones that receive the highest quantity of precipitation as well as the complex orography of the study area. The presence of the Andes mountain range directly influences precipitation, and the present study area corresponds with one

of the lowest sectors of the mountain range known as the inter-Andean Depression (Richter and Moreira, 2005). In a study, Cuervo-Robayo et al. (2014) developed climate surfaces for Mexico using the thin-plate smoothing method, and the results





were generally satisfactory. However, the authors similarly indicated that high standard error values were mostly found for mountain ranges.

In another study in an Andean basin of Argentina, Pessacg et al. (2015) evaluated different precipitation data sets with distinct spatial and temporal resolutions; the calibration of the model varied per precipitation database. Notably, these

authors found that precipitation errors of ± 30% led to errors in water yield of 50% to 150% (-45% to -60%) in some sub-basins.

According to Hamel and Guswa (2015), Z values lower than 5 are unlikely. In the present study, despite the calibration of the water yield model in the five basins being satisfactory, three of the basins had low Z values, confirming that precipitation was underestimated. It was also determined that lower values of Z corresponded with higher values of water yield and,

similarly, that water yield decreased to the extent that Z values increased (Figure 6). The studies of Pessacg et al. (2015) and Zhang et al. (2012) confirm this relationship.

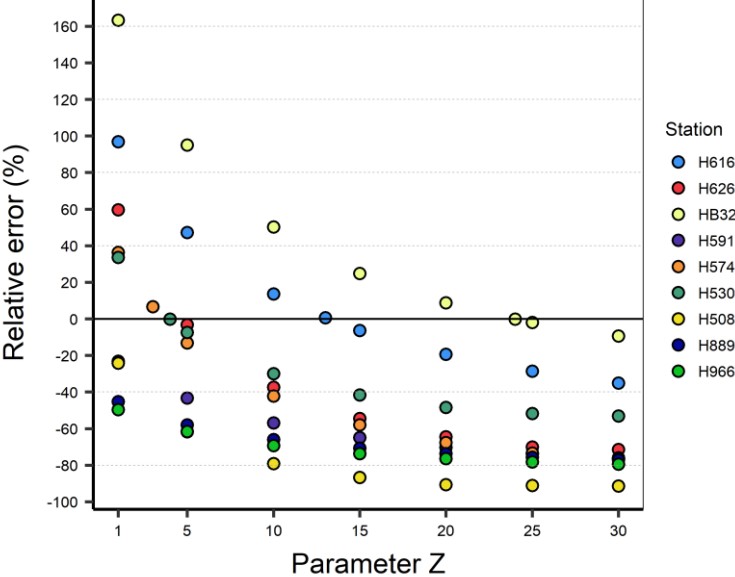

**Figure 6: Calibration of the model as a function of ecohydrological parameter Z.**

**Water yield**

The basins located in the Amazon region had the highest water production. These basins are influenced by masses of hot air with high humidity proceeding from evaporative processes in wetlands and evapotranspiration in rain forests, which both generate significant precipitation through processes of adiabatic cooling (Pourrut et al., 1995). Overall, in this region, precipitation is uniformly distributed throughout the year. However, in the Mayo basin, there is a slight decrease in precipitation from the months of June to October.

Meanwhile, the basins located in the Andean mountains and valleys to the southwest of the mountain chain are generally wet in the high parts and dry in the low parts as a result of the Foehn effect (Emck, 2007). A clear example of this phenomenon





occurs in the Valley of Catamayo (northeast of the Catamayo basin), which has a completely arid climate as a result of the Foehn effect and easterly winds. The driest months are June to August, which have very little or no precipitation. Meanwhile, the short wet season is characterized by intense rains that arrive with humid air masses originating from the South. Additionally, the Valley of Vilcabamba (southeast of the Catamayo basin) show a similar pattern. However, its

climate is semi-arid (with a similar seasonality) because of its proximity to the Cordillera Real and the slightly weaker Foehn effect (Rollenbeck and Bendix, 2011).

Another study in the southern region of Ecuador (Aguirre et al., 2015) evaluated water production as an ecosystem service in ZP7. Specifically, average water production was calculated seasonally (for the rainy and dry season) using a run-off equation. The results showed greater water production in the northwestern zone (Zamora basin) and lower production in the

coastal zones and Andean valleys, which is in agreement with the results of the present study. Therefore, in the present study, the zones of high and low water production were directly influenced by the distribution of precipitation; however, the influence of land cover was also evidenced, as the model captured variation in water yield with respect to LUC (Table 5).

The main limitations of the InVEST water yield model are its inability to consider seasonal or sub-seasonal variability as well as infrastructure for ground water or for redistributing water flow. However, the model is deterministic and is based on

simplifications of widely accepted hydrological relationships. It has the advantage of being based on relatively simple code that users can understand and adjust as necessary (Vigerstol and Aukema, 2011). Based on this simplified representation of the hydrological process, the InVEST Water Yield model can be used to quantify and map related ecosystem services relatively rapidly (Lüke and Hack, 2018).

## 5 Conclusions

The modeling of water yield in the majority of hydrographic basins was satisfactory. The model allowed the order of importance of the basins in terms of water production to be determined despite underestimating precipitation. The basins with the highest water production were Zamora and Mayo, highlighting the importance of these basins in the study area.

However, because water yield was underestimated in some of the hydrographic basins of the study area, it is necessary to explore different algorithms for estimating precipitation and temperature at different spatial and temporal resolutions in order

to determine the most ideal method for evaluating water yield in the southern region of Ecuador.

In the present study, point climate data were used and subsequently interpolated. These data mostly originated from meteorological stations located in the low parts of valleys. For this reason, to generate more accurate estimations of annual water yield in the study area, it is necessary to increase the density of meteorological stations and to improve the spatial distribution of these stations.

Different Budyko approaches, including the framework used in the present study, have been developed for calculating long-term water balance (Hamel and Guswa, 2015). Further research studies should examine the effects of land use and cover changes on water production as well as the effects of climate changes in future projections.



In conclusion, the obtained results are an initial estimate of the water yield at the hydrographic basin level of the southern region of Ecuador. The results may have applications in different fields, including water supply services, hydroelectric energy production, in the identification of protection areas and the mitigation of water risk. In particular, these results can be used to evaluate the impacts of land use changes and climate change and to thereby facilitate the planning and local management of water resources in the long term.

**Author Contribution:** Conceptualization, S.M.L., M.A.G.A. and K.M.B.; methodology, S.M.L., M.A.G.A., K.M.B. and L.B.; software, L.R.M.S. and A.P.C.R.; formal analysis, S.M.L., C.A.M.L. and M.A.G.A.; investigation, S.M.L. and M.A.G.A.; data curation, L.B., A.P.C.R. and L.R.M.S; writing—original draft preparation, K.M.B. and M.A.G.A.; writing—review and editing, A.P.C.R., L.R.M.S. and C.A.M.L.

**Competing interests:** The authors declare that there are no conflicts of interest.

**Acknowledgments:** This study was carried out with the financial support provided by CONACYT under research projects 248553 and the UAEM/Quebec Grant 4192/2016E.

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
