# Peer review of "Estimation of water yield in the hydrographic basins of southern Ecuador"

_Hydrology and Earth System Sciences, 2018_

## Referee Comment (RC1) · Anonymous Referee #1 · 11 Nov 2018

General comment

Estimating water yields in poorly gauged areas with large topographical and climate heterogeneity remains challenging. Therefore, the combination of multiple data sources and the use of the model to solve the water budget in areas like the south of Ecuador is very interesting. However, the manuscript is more focused on the specific case than in the broader implications of the proposed methods. This is evidenced in the lack of clear research questions that could go beyond the current objective: "to estimate and map annual water yield for the 1970-2015 period in nine hydrographic basins of the south of Ecuador". By stating the research questions the conclusions could also be stronger and more relevant to a broader public. Furthermore, you have a large data set (1970-2015) why not use part of it to calibrate the model and the other to validate?

Specific comments

Pg3L2: "However, the uncertainty introduced by errors in climate data can be significant and non-spatially heterogeneous, subsequently affecting the estimated spatial distribution of water yield." How will this be considered? does it need to be considered?

Pg3L8 in relation to pg2L5. Water yields estimated for the 1970-2015 time period are not necessarily "current" given the fast changes caused by land cover and climate change. This also raises the question of how much hydrological variability was observed within this long time period? Could any trends be found? This could be outside the scope of your paper; however, I believe it is necessary to describe the observed behavior of the hydro-climatic variables within the study period as context information.

Pg3L11: Are you sure the precipitation range over the Amazon side of the Andes is that low? Please see precipitation data for the cerro El Consuelo (> 4000 mm) Bendix, J., Rollenbeck, R., Fabian, P., Emck, P., Richter, M., Beck, E., 2008. Climate Variability: temporal heterogeneities, In: Beck, E., Bendix, J., Kottke, I., Makeschin, F., Mosandl, R. (Eds.), Gradients in a Tropical Mountain Ecosystem of Ecuador, Ecological Studies. Springer Berlin Heidelberg, Berlin, Heidelberg, pp. 281–290. doi:10.1007/978-3-540-73526-7 Why not include the data from El Consuelo, if it is available?

pg7L9: You are using a land use map of 2014, which is at the end of the study period (1970-2015). It is reasonable to assume that significant land-cover change has taken place during the study period. How can you assume that a 2014 map is representative for the study period? If this assumption is not supported, how can it affect your results?

pg7L16: You estimated PAWC from data available from HWSD? or you obtained PAWC data from HWSD? if the first, please state how.

pg8L8: Sorry, I do not understand what do you mean by "The calibration was performed up to the elevation of the stations".

pg10L4 and Table 4: Your study area has a very sharp precipitation gradients accord-

ing to the position within the Andean system (e.g. Amazonian, High-Andean valleys, Pacific slopes). It can be easily predicted that the basins in the high Andean valleys will have a lower water yield and that the ones from the Amazonian side will have a larger water yield by just using precipitation data. In this regard, your aim could go beyond stating which basins have higher water yields and maybe focus more on the limitations to estimate water yields and the importance of understanding how other variables different to precipitation modulate them. Can you please provide information on the position within the Andean system of each studied basin. This information can be provided by an underlying position classification in figure 3f.

pg13Figure6: This is part of your results and they could be better placed in the results section. Also, I do not see how the plot shows what you are trying to explain in L9-10 because in the plot you cannot see anything related to water yield. This plot can be improved. For example, the colors representing the basins could be graduated from higher to lower water yield

pg14L24: It is not a matter of algorithms, it is a problem of the available data. If the gauging density does not represent the spatial heterogeneity of rainfall there is no algorithm that can fix it. Once we have data to represent the spatial variability then we can evaluate the best way to interpolate it.

Technical corrections

pg6L7: An image is not the same as a map. I think you are referring to precipitation and temperature distribution maps

pg10L10 Where are you getting these values from? include reference to Table 5

pg11 and Table5: Please correct the units Mm3.

pg14L12: I could not find in Table 5 any reference to LUC
* * *
529, 2018.

---

## Short Comment (SC1) · 14 Nov 2018

The manuscript by Minga-León and coauthors provides an estimation of water production in nine hydrographic basins in southern Ecuador. Without offering a complete review of the article, which would be given during the peer-review process, I was surprised to see that the literature review, especially the one offered in the Introduction, presents references that date back to 2015 at the latest. Apart from Redhead et al (2016) and Li et al. (2018), both referring to the InVEST model, there are not updated citations on tropical, Andean, and Amazonian hydrology that can complement, contextualise, and offer further discussion to enrich this work.

I provide here six relevant articles that the authors could read, include, and use as a

starting point to strengthen their literature review and scientific content for their study:

- Manz, B., Páez-Bimos, S., Horna, N., Buytaert, W., Ochoa-Tocachi, B. F., Lavado-Casimiro, W., and Willems, B. (2017) Comparative Ground Validation of IMERG and TMPA at Variable Spatio-temporal Scales in the Tropical Andes. J. Hydrometeorol., 18: 2469–2489. doi: 10.1175/JHM-D-16-0277.1

- Ochoa-Tocachi, B. F., W. Buytaert, and B. De Bièvre (2016), Regionalization of land-use impacts on streamflow using a network of paired catchments, Water Resour. Res., 52: 6710–6729. doi: 10.1002/2016WR018596.

- Ochoa-Tocachi, B. F., Buytaert, W., De Bièvre, B., Célleri, R., Crespo, P., Villacís, M., Llerena, C. A., Acosta, L., Villazón, M., Guallpa, M., Gil-Ríos, J., Fuentes, P., Olaya, D., Viñas, P., Rojas, G., and Arias, S. (2016) Impacts of land use on the hydrological response of tropical Andean catchments. Hydrol. Process., 30: 4074–4089. doi: 10.1002/hyp.10980.

- Ochoa-Tocachi, B. F., Buytaert, W., Antiporta, J., Acosta, L., Bardales, J. D., Célleri, R., Crespo, P., Fuentes, P., Gil-Ríos, J., Guallpa, M., Llerena, C., Olaya, D., Pardo, P., Rojas, G., Villacís, M., Villazón, M., Viñas, P., and De Bièvre, B. (2018) High-resolution hydrometeorological data from a network of headwater catchments in the tropical Andes. Sci. Data, 5: 180080. doi: 10.1038/sdata.2018.80

- Timbe, E., Feyen, J., Timbe, L., Crespo, P., Célleri, R., Windhorst, D., Frede, H.-G., and Breuer, L. (2017) Multicriteria assessment of water dynamics reveals subcatchment variability in a seemingly homogeneous tropical cloud forest catchment. Hydrol. Process., 31: 1456–1468. doi: 10.1002/hyp.11146

- Zulkafli, Z., Buytaert, W., Manz, B., Véliz Rosas, C., Willems, P., Lavado-Casimiro, W., Guyot, J.-L., and Santini, W. (2016) Projected increases in the annual flood pulse of the Western Amazon. Environ. Res. Lett., 11: 014013. doi: 10.1088/1748-9326/11/1/014013

---

## Author Comment (AC1) · 26 Dec 2018

Dear reviewer: We are thankful for the time taken to review our manuscript, and we consider that the questions and comments are appropriate. These have contributed to substantially improving our work, and we hope to make the necessary changes.

General comment: Estimating water yields in poorly gauged areas with large topographical and climate heterogeneity remains challenging. Therefore, the combination of multiple data sources and the use of the model to solve the water budget in areas like the south of Ecuador is very interesting. However, the manuscript is more focused on the specific case than in the broader implications of the proposed methods. This is evidenced in the lack of clear research questions that could go beyond the current

objective: "to estimate and map annual water yield for the 1970-2015 period in nine hydrographic basins of the south of Ecuador". By stating the research questions the conclusions could also be stronger and more relevant to a broader public. Furthermore, you have a large data set (1970-2015) why not use part of it to calibrate the model and the other to validate?

Response: It is important to mention that the annual water yield model is designed for modeling averages in the long term and not in the short term or annually. Therefore, annual precipitation and reference evapotranspiration (ET0) should represent averages over the long term, preferably over at least 10 years according to the recommendations of Sharp et al. (2018). However, to calibrate and validate the model, hydrometric data (water flow) are necessary. For our study, the extent of the water flow data only allowed us to calibrate the model based on the Z parameter, an empirical constant that captures the effects of plant cover, climate, and topography.

Specific comments

1) Comment: Pg3L2: "However, the uncertainty introduced by errors in climate data can be significant and non-spatially heterogeneous, subsequently affecting the estimated spatial distribution of water yield." How will this be considered? does it need to be considered?

Response: The above mentioned question is important. In our study, to minimize uncertainty as a result of errors in climate data, a data quality analysis was carried out. Atypical values were detected, which were subsequently validated, corrected, or eliminated according to various criteria (Pg8L11). Summary of data detected as atypical:

Precipitation: Eliminated 17, Validated 23 and Corrected 2. Maximum temperature: Validated 2. Total data analyzed 44.

However, this study did not specifically focus on an analysis of uncertainty as a result

of errors in climate data. Previous studies such as those of Hamel and Guswa (2015), Redhead et al. (2016), and Sánchez-Canales et al. (2012) were also considered. These studies identified uncertainty stemming from variability in climate data and the empirical variables in the corresponding models. Their results show a high sensitivity to precipitation and, to a lesser extent, to evapotranspiration data. The study of Pessacg et al. (2015) showed that errors in precipitation of ±30% led to errors in water yield of 50% to 150% (−45% to −60%) in some sub-basins. Meanwhile, the sensitivity of the empirical variable of the model (parameter Z) is specific to each basin since its effect on yield is modulated by precipitation and available water content (AWC) for plants.

2) Comment: Pg3L8 in relation to pg2L5. Water yields estimated for the 1970-2015 time period are not necessarily "current" given the fast changes caused by land cover and climate change. This also raises the question of how much hydrological variability was observed within this long time period? Could any trends be found? This could be outside the scope of your paper; however, I believe it is necessary to describe the observed behavior of the hydro-climatic variables within the study period as context information.

Response: We agree, it is not necessarily the current water yield considering the land use/cover (LUC) map. The climate variables represent the annual averages for the 1970–2015 period, whereas the LUC characteristics are representative of the analyzed period. The most recent LUC map for the study period (2013–2014) was used. In this study, it would be difficult to obtain hydrological variability in the short term, mostly because of the lack of information in the region in regard to the physical characteristics of soil cover as well as observed water flow data. In this sense, changes were made in the manuscript to clarify these concepts concerning the temporality of data. 1. A clarification was included in the introduction (Pg3L13) that water yield is estimated considering the LUC map of the 2013–2014 period. Also, the word "current" was eliminated at Pg2L5. 2. Figure 2 was included (Pg5) to show the tendencies over time in the hydroclimatic data of the hydrographic basins.

3) Comment: Pg3L11: Are you sure the precipitation range over the Amazon side of the Andes is that low? Please see precipitation data for the cerro El Consuelo (> 4000 mm) Bendix, J., Rollenbeck, R., Fabian, P., Emck, P., Richter, M., Beck, E., 2008. Climate Variability: temporal heterogeneities, In: Beck, E., Bendix, J., Kottke, I., Makeschin, F., Mosandl, R. (Eds.), Gradients in a Tropical Mountain Ecosystem of Ecuador, Ecological Studies.Springer Berlin Heidelberg, Berlin, Heidelberg, pp. 281–290. doi:10.1007/978-3-540- 73526-7 Why not include the data from El Consuelo, if it is available?

Response: Surface precipitation in this study was based on the average values for the 1970–2015 period. Stations located in the Cerro El Consuelo were used in addition to precipitation data from a total of 160 meteorological stations distributed across the study area. Of these, 115 stations belong to the National Meteorology and Hydrology Institute of Ecuador (Instituto Nacional de Meteorología e Hidrología de Ecuador [INAMHI]), 32 to the National Meteorology and Hydrology Service of Peru (Servicio Nacional de Meteorología e Hidrología de Perú [SENAMHI]), and 13 to the San Francisco Scientific Station (Estación Científica San Francisco [ECSF]). The stations of the ECSF are generally located above 2000 masl and distributed across Cerro El Consuelo, Cajanuma, El Tiro, and Tapichachalaca, among other sites. However, the data from these stations are relatively short (from no more than 5 years ago).

4) Comment: pg7L9: You are using a land use map of 2014, which is at the end of the study period (1970-2015). It is reasonable to assume that significant land-cover change has taken place during the study period. How can you assume that a 2014 map is representative for the study period? If this assumption is not supported, how can it affect your results?

Response: Changes in land use/cover (LUC) during the study period could modify the results because of the influence of the physical characteristics of plants. However, the results of this study are valid for the average climate variables (1970–2015) considering the official LUC map of Ecuador for the 2013–2014 period. In future studies, it is important to generate more precise data on variability in water yield with respect to

different LUCs if enough temporal data on observed water flows can be obtained.

5) Comment: pg7L16: You estimated PAWC from data available from HWSD? or you obtained PAWC data from HWSD? if the first, please state how.

Response: We appreciate the comment, there was some confusion in the explanation, which was corrected on Pg9L13. The AWC values (mm) were obtained from the HWSD database, and these values were divided by the minimum value of the root restriction depth or rooting depth of vegetation (mm) with the goal of obtaining the required fraction (PAWC) by the Water Yield model. The PAWC values are dimensionless (0 to 1) and are basically obtained by solving equation 5 in the document.

PAWC=AWC/(Min (Rest.layer.depth,root.depth))

6) Comment: pg8L8: Sorry, I do not understand what do you mean by "The calibration was performed up to the elevation of the stations".

Response: Thank you, the phrase was eliminated for not being clear after confirming that the idea was already mentioned in the previous paragraph, which refers to the performance of the calibration of parameter Z by hydrographic unit up to the location of the hydrometric stations and not at the outlet of each hydrographic unit.

7) Comment: pg10L4 and Table 4: Your study area has a very sharp precipitation gradients according to the position within the Andean system (e.g. Amazonian, High-Andean valleys, Pacific slopes). It can be easily predicted that the basins in the high Andean valleys will have a lower water yield and that the ones from the Amazonian side will have a larger water yield by just using precipitation data. In this regard, your aim could go beyond stating which basins have higher water yields and maybe focus more on the limitations to estimate water yields and the importance of understanding how other variables different to precipitation modulate them. Can you please provide information on the position within the Andean system of each studied basin. This information can be provided by an underlying position classification in figure 3f.

Response: Correct, it is known that the climate variables (precipitation and evapotranspiration) modulate the availability of water yield (Hamel and Guswa, 2015; Pessacg et al., 2015; Sánchez-Canales et al., 2012). For this reason, in this study, the calibration of the Z parameter was the focus. This parameter considers soil physical characteristics and plant cover in the study area, which are directly related with the $\omega$ parameter. With respect to the comment on Figure 3f, we decided to modify the map of the study area in order to show the influence of the Andean system on the hydrographic basins but did not modify the suggested figure given that it only refers to the model inputs.

8) Comment: pg13Figure6: This is part of your results and they could be better placed in the results section. Also, I do not see how the plot shows what you are trying to explain in L9-10 because in the plot you cannot see anything related to water yield. This plot can be improved. For example, the colors representing the basins could be graduated from higher to lower water yield.

Response: Thank you, we decided to locate the calibration graph of the Z parameter in the Results section.

9) Comment: pg14L24: It is not a matter of algorithms, it is a problem of the available data. If the gauging density does not represent the spatial heterogeneity of rainfall there is no algorithm that can fix it. Once we have data to represent the spatial variability then we can evaluate the best way to interpolate it.

Response: Correct, it is for this reason that a reference (Pg17L23) is made to the lack of available climate data, which is the main problem in the estimation of water yield. In this study, an effort was made to compile the greatest quantity of available data in the study area for the 1970–2015 period. The utilized interpolation method was thin-plate smoothing (Hutchinson, 2006).

Technical corrections:

Pg8L4: An image is not the same as a map. I think you are referring to precipitation

and temperature distribution maps. Corrected, the word "images" was corrected and replaced with "maps." pg10L10 Where are you getting these values from? include reference to Table 5. Corrected. pg11 and Table5: Please correct the units Mm3. A note was added at the foot of the table to explain the units (pg13). pg14L12: I could not find in Table 5 any reference to LUC. A brief explanation was added on pg17L3 about the influence of LUCs on variation in water yield.

Note. The corrected manuscript is attached as a supplement.

Please also note the supplement to this comment:
https://www.hydrol-earth-syst-sci-discuss.net/hess-2018-529/hess-2018-529-AC1-supplement.pdf

————————————————

[Figure]

**Fig. 1.** Location of the study area
Interactive
comment

**Fig. 2.** Hydroclimatic variables

**Supplement:**

**Estimation of water yield in the hydrographic basins of southern Ecuador**

Saula Minga-León [1], Miguel Angel Gómez-Albores [1], Khalidou M. Bâ [1], Luis Balcázar [1], Luis Ricardo Manzano-Solís [2], Angela P. Cuervo-Robayo [3,4], Carlos Alberto Mastachi-Loza [1]

[1]Instituto Interamericano de Tecnología y Ciencias del Agua, Universidad Autónoma del Estado de México, Toluca 50200, México
[2]Facultad de Geografía, Universidad Autónoma del Estado de México, Toluca 50110, México
[3]Comisión Nacional para el Conocimiento y Uso de la Biodiversidad (Conabio), Ciudad de México 14010, México
[4]Departamento de Zoología, Instituto de Biología, Universidad Nacional Autónoma de México, Ciudad de México 04510, México

*Correspondence to*: Miguel Angel Gómez-Albores (magomeza@uaemex.mx)

**Abstract.** Humans greatly benefit from natural water resources, also known as hydrological ecosystem services. However, these services may be reduced by population growth, land use changes, and climate change. As these problems become more critical, the need to quantify water resources increases. The estimation of water yield and its distribution are of great importance for the management of water resources. In the present study, the average annual water yield of the hydrographic basins in the southern region of Ecuador was estimated for the 1970–2015 period using the InVEST water yield model based on the Budyko framework. The model estimates annual surface run-off at the pixel, sub-basin, and basin level considering the following variables: precipitation, actual evapotranspiration, land cover/use, soil depth, and available water content for plants. The model was calibrated by varying the ecohydrological parameter $Z$ to reduce error between estimated and observed water yield. The results showed that the modeling of water yield in the majority of the hydrographic basins was satisfactory, allowing the basins to be ranked according to their importance for water production. The Mayo and Zamora basins had the highest water production, corresponding with 934 and 1218 mm year$^{-1}$, respectively, while the Alamor and Catamayo basins had the lowest water production, corresponding with 206 and 291 mm year$^{-1}$, respectively. The present study provides an initial estimate of water yield at the basin level in the southern region of Ecuador, and the results can be used to evaluate the impacts of land cover changes and climate change over time.

**1 Introduction**

Humans receive a variety of benefits from natural water resources, also known as hydrological ecosystem services (HESs). Several of the main HESs include regulation of the water cycle, high water yields, maintenance of water quality, and aquifer recharge. These services can be negatively affected if natural ecosystems are converted to other land uses by human activities (Quintero, 2010). Decades of anthropogenic intervention have altered land covers and uses, resulting in the conversion of natural lands to pastures, crops, deforested areas, urban constructions, and mining operations, among other land uses (Crespo

et al., 2014). As these problems become more critical, effective land use planning is necessary to guarantee the continued provision of HESs. Such planning requires a good understanding of hydrology and can also be used to support key decisions about the management of water resources in the near future (Buytaert et al., 2006b). The uncertainty associated with climate change and its impacts on the water cycle and water availability further incentivize the need to generate detailed inventories
5  of water resources (Tallis and Polasky, 2009).

In Ecuador, few studies have quantified water resources at the hydrographic basin level (Buytaert et al., 2006b; Duque Yaguache and Vázquez Zambrano, 2015). The main challenges for the advancement of hydrological knowledge have been the difficulty of implementing and maintaining observation networks in complex and remote environments and the lack of recognition of ecosystems as water suppliers (Célleri et al., 2009). However, some microbasins have been studied in the wet
10  paramos and tropical mountains of southern Ecuador. Of these studies, several can be highlighted, whose main contributions have been the conceptualization of hydrological processes (Célleri et al., 2009; Célleri and Feyen, 2009; Llambí et al., 2012), water modeling (Balcázar et al., 2016; Buytaert et al., 2006a; Oñate and Aguilar, 2003; Vázquez, 2015), monitoring and quantification of the main climate variables (Campozano et al., 2016; Ochoa-Tocachi et al., 2018), documenting climate variability (Bendix et al., 2008; Buytaert et al., 2006c; Celleri et al., 2007; Fries et al., 2014; Luna-Romero et al., 2018;
15  Rollenbeck and Bendix, 2011), and determining the effect of land use changes on water behavior (Buytaert et al., 2005, 2007; Crespo et al., 2014; Ochoa-Tocachi et al., 2016) as well as the impacts of climate change on hydrology (Breuer et al., 2013; Buytaert et al., 2009, 2010; Buytaert and Bievre, 2012; Mora et al., 2014). Generally, these studies have been developed at small scales over relatively short time frames.

The importance of studying Andean basins rests on their importance as the main providers of water for the Andes highlands
20  as well as for the adjacent, extensive lowlands and a portion of the coastal plains (Buytaert et al., 2006b). Crespo et al. (2014) highlighted that water production in paramo microbasins in their natural state is around 75% of precipitation in terms of quantity. The high water production of these basins is due to the high quantity of precipitation as well as the low water consumption of vegetation. However, the urban growth and concentration of agriculture in the inter-Andean valleys have directly influenced the increase in the demand for water for different uses.

[revised manuscript text omitted]

30   Moreira, 2005).

[Figure]

**Figure 1: Location of the study area in relation to (a) Southern America, (b) Ecuador and the Andean system. (c) Location of the hydrometric stations in relation to the hydrographic basins.**

5    The hydroclimatic variables analyzed in the 1970–2015 period showed that precipitation in the hydrographic basins, except for the Zamora and Mayo basins, presents high spatial-temporal variability (Figure 2

Figure 2: Magnitude and evolution of the hydroclimatic variables per hydrographic basin for the 1970–2015 period.  and Figure 3a). The precipitation level varied between ~300 and ~1800 mm year$^{-1}$. However, in years with El Niño events, the precipitation

10   level reached ~3400 mm year$^{-1}$. In the Zamora and Mayo basins (Figure 2) located along the Amazon slope, precipitation varied between ~1000 and ~2200 mm year$^{-1}$. The mean temperatures of the hydrographic basins (Figure 2) along both slopes varied between 14 to 25 °C. At the spatial level, areas near the Andes mountain ranges were colder, whereas the coastal areas were warmer (Figure 3b).

[Figure]

**Figure 2: Magnitude and evolution of the hydroclimatic variables per hydrographic basin for the 1970–2015 period.**

[Figure]

[revised manuscript text omitted]
 fraction (PAWC)**: This parameter was generated based on the AWC and the minimum value of the soil depth or root depth of vegetation (Equation (5)). The AWC and soil depth were obtained from the HWSD databases
15    (FAO et al., 2012), and the root depth was obtained from Allen et al. (1998) and Sharp et al. (2018). The result was a raster image of PAWC with values between 0 and 1 (dimensionless).

**Biophysical attribute table**: A table containing biophysical attributes for each LUC class (Table 1) was generated in text format (.csv). The attributes are as follows: Id, LUC class, evapotranspiration coefficient ($K_c$), root depth, and vegetation presence. The values for $K_c$ and root depth were taken from Allen et al. (1998) and Sharp et al. (2018). $K_c$ is used to obtain
20    potential evapotranspiration from $ET_0$; the value ranges from 0 to 1.5. The field vegetation presence determines which equation will be used to estimate actual evapotranspiration (AET) in the WY model: A value of 1 corresponds with LUCs with vegetation (Eq. 2), and a value of 0 corresponds with LUCs without vegetation (Eq. 6).

**Ecohydrological parameter $Z$**: $Z$ is an empirical constant with a value of 1 to 30 that was determined during the calibration, which involved the comparison of estimated data with observed data. At following, the calibration process is described in
25    greater detail.

**Table 1: Biophysical attribute table used to execute the water yield model.**

| Id | LUC classes | $K_c$ | Root depth (mm) | Presence vegetation |
|----|-------------|-------|-----------------|---------------------|
| 1 | Native forest | 1 | 5000 | 1 |
| 2 | Natural water body | 1 | - | 0 |
| 3 | Semi-permanent crop | 0.6 | 1000 | 1 |
| 4 | Grassland | 0.7 | 1000 | 1 |
| 5 | Permanent crop | 0.6 | 1000 | 1 |
| 6 | Shrub vegetation | 0.5 | 2000 | 1 |
| 7 | Agro-livestock mosaic | 0.6 | 1000 | 1 |
| 8 | Populated area | 0.3 | - | 0 |
| 9 | Area without vegetation | 0.3 | - | 0 |
| 10 | Artificial water body | 1 | - | 0 |
| 11 | Paramo | 0.7 | 1500 | 1 |
| 12 | Herbaceous vegetation | 0.7 | 1500 | 1 |
| 13 | Forest plantation | 1 | 5000 | 1 |
| 14 | Annual crop | 0.6 | 900 | 1 |
| 15 | Infrastructure | 0.3 | - | 0 |

**2.4 Calibration of the Water Yield model**

To calibrate the WY model, as stated above, the ecohydrological parameter *Z* and observed data (long-term average water flow) collected from nine hydrological stations were used. Unfortunately, hydrometric stations were not present at the outlets of the hydrographic basins; data from these stations are ideal for model calibration. The nine available hydrometric stations were located in sub-basins of the hydrographic basins of the study area (Figure 1 and Figure 2).

The calibration was performed by varying the Z value from 1 to 30 until the error was minimized to the greatest extent possible. In this case, the error is determined by the difference between estimated and observed water yield. Observed water yield was determined from data on daily water flow from INAMHI. The calibration period of each hydrographic sub-basin depended on the availability of observed data from hydrometric stations (Table 2).

**Table 2: Information on hydrometric stations and calibration period for the water yield model.**

| Station code | Hydrometric station | Hydrographic basin | Calibration period | Evaluated years | Excluded years |
|---|---|---|---|---|---|
| H0616 | Alamor at Saucillo (Dj Celica) | Alamor | 1970–2011 | 42 | - |
| H0626 | Macará at Pte. Internacional | Macará | 1979–1994 | 16 | - |
| HB32 | Catamayo at Vicin | Catamayo | 2006–2014 | 8 | 2012 |
| H0591 | Puyango at Cpto. Militar (Pte. Carretera) | Puyango | 1970–2011 | 42 | 1995 |
| H0574 | Arenillas at Arenillas | Santa Rosa | 1970–2011 | 34 | 1984-1990 |
| H0530 | Jubones at Ushcurrumi | Jubones | 1970–2011 | 42 | - |
| H0508 | Chaguana at Pte. Carretera | Pagua | 2003–2011 | 9 | - |
| H0889 | Zamora DJ Sabanilla (at Zamora) | Zamora | 1982–2011 | 28 | 1985, 2002 |
| H0966 | Mayo AJ Qda. Zumbayacu | Mayo | 1982–2011 | 15 | 1984, 1986-1988, 1993-1995, 1997-2002, 2005-2006 |

The units for the observed data were cubic meters per second ($m^3/s$), and the units of the estimated data were millimeters per year (mm year$^{-1}$). Accordingly, the observed data were transformed to mm year$^{-1}$.

5    In the hydrographic sub-basins, the diversion of water flow by irrigation systems was additionally considered. Only water flows that could influence the calibration of the model were taken into account (Table 3), such as the Zapotillo irrigation system located before station HB32 that diverts approximately 8 $m^3/s$. Small water channels that communities use for irrigation were not considered because information was lacking on the diverted water volume and because the diverted quantities were considered minimal.

10    **Table 3: Irrigation systems considered in the calibration of the water yield model.**

| Hydrographic basin | Station code | Irrigation channels | Total diverted volume ($m^3/s$) | Source | Date |
|---|---|---|---|---|---|
| Macará | H0626 | 7 | 3.29 | GADP-Loja (2013) | up to 2012 |
| Catamayo | HB32 | 7 | 9.40 | | |

The calibrated *Z* values were used to obtain the water yield of each hydrographic basin under the assumption that the calibrated value captured the climate characteristics, precipitation intensity, and topography of the basins.

**3 Results**

**3.1 Calibration of the water yield model**

The calibration of the model provided an analysis of the sensitivity of water yield with respect to ecohydrological parameter Z (Figure 5). The extreme values of Z, 1 and 30, led to different simulations of water yield. For example, in the case of the Alamor in Saucillo (H0616) station, the Z values of 1 and 30 led to errors of 95% and 35%, respectively, with respect to the Z value of 13 at which the error was minimized to 0.58%. The lowest Z values had higher errors in the simulations of water yield than the highest Z values.

[revised manuscript text omitted]

**4 Discussion**

**Calibration of the Water Yield model**

The Z empirical parameter remains uncertain according to several studies (Hamel and Guswa, 2015; Pessacg et al., 2015; Redhead et al., 2016). High Z values correspond with areas that receive a greater number of rainfall events per year, whereas low Z values indicate few rainfall events per year. However, in the present study, despite the satisfactory calibration of five hydrographic basins, stations H016 (Jubones), H0626 (Macará), and H0574 (Santa Rosa) were calibrated with low Z values, indicating that these zones are arid, even though precipitation occurs during the entire year in the high part of the Jubones basin. This can be attributed to the forcing of the Z parameter with the objective of minimizing the error between simulated and observed water yield.

The high variation of calibrated basins in the Z parameter (3 to 24) can be attributed to the unique characteristics of the basins in regard to plant cover, soil properties, and topography despite their proximity. The w variable modulates these characteristics and is conditioned by the AWC-precipitation relationship and the empirical factor Z. This latter factor captures the intensity of precipitation and the topography of the basin, which are not described by available water content (AWC) and annual precipitation.

Water yield was underestimated for the stations that were unable to be calibrated. This is possibly due to the precipitation data, as precipitation is the variable most sensitive to the estimation of water yield according to several previous studies (Goyal and Khan, 2016; Hamel and Guswa, 2015; Redhead et al., 2016; Sánchez-Canales et al., 2012). Notably, Rollenbeck and Bendix (2011) evaluated a set of local area weather radar (LAWR) images in southern Ecuador and found that precipitation reached

5    4000 mm in the high mountains. However, the images obtained in the present study from the interpolation indicated a precipitation level of 2400 mm for this same area.

As previously mentioned, the underestimation of precipitation is possibly due to the low density of meteorological stations in the high mountain zones that receive the highest quantity of precipitation as well as the complex orography of the study area. The presence of the Andes mountain range directly influences precipitation, and the present study area corresponds with one

10    of the lowest sectors of the mountain range known as the inter-Andean Depression (Richter and Moreira, 2005). In a study, Cuervo-Robayo et al. (2014) developed climate surfaces for Mexico using the thin-plate smoothing method, and the results were generally satisfactory. However, the authors similarly indicated that high standard error values were mostly found for mountain ranges.

In another study in an Andean basin of Argentina, Pessacg et al. (2015) evaluated different precipitation data sets with distinct

15    spatial and temporal resolutions; the calibration of the model varied per precipitation database. Notably, these authors found that precipitation errors of ± 30% led to errors in water yield of 50% to 150% (-45% to -60%) in some sub-basins.

**Water yield**

The basins located in the Amazon region had the highest water production. These basins are influenced by masses of hot air with high humidity proceeding from evaporative processes in wetlands and evapotranspiration in rain forests, which both

20    generate significant precipitation through processes of adiabatic cooling (Pourrut et al., 1995). Overall, in this region, precipitation is uniformly distributed throughout the year. However, in the Mayo basin, there is a slight decrease in precipitation from the months of June to October.

Meanwhile, the basins located in the Andean mountains and valleys to the southwest of the mountain chain are generally wet in the high parts and dry in the low parts as a result of the Foehn effect (Emck, 2007). A clear example of this phenomenon

25    occurs in the Valley of Catamayo (northeast of the Catamayo basin), which has a completely arid climate as a result of the Foehn effect and easterly winds. The driest months are June to August, which have very little or no precipitation. Meanwhile, the short wet season is characterized by intense rains that arrive with humid air masses originating from the South. Additionally, the Valley of Vilcabamba (southeast of the Catamayo basin) show a similar pattern. However, its climate is semi-arid (with a similar seasonality) because of its proximity to the Cordillera Real and the slightly weaker Foehn effect (Rollenbeck and

30    Bendix, 2011).

Another study in the southern region of Ecuador (Aguirre et al., 2015) evaluated water production as an ecosystem service in ZP7. Specifically, average water production was calculated seasonally (for the rainy and dry season) using a run-off equation. The results showed greater water production in the northwestern zone (Zamora basin) and lower production in the coastal

zones and Andean valleys, which is in agreement with the results of the present study. However, the influence of vegetation cover was also evidenced: The model was able to capture variation in water yield with respect to LUC. Precipitation and $ET_o$ in some basins were similar, whereas the AET was different (Table 5) as a consequence of the influence of the $K_c$ of each LUC class in the estimation of AET.

5 The main limitations of the InVEST water yield model are its inability to consider seasonal or sub-seasonal variability as well as infrastructure for ground water or for redistributing water flow. However, the model is deterministic and is based on simplifications of widely accepted hydrological relationships. It has the advantage of being based on relatively simple code that users can understand and adjust as necessary (Vigerstol and Aukema, 2011). Based on this simplified representation of the hydrological process, the InVEST Water Yield model can be used to quantify and map related ecosystem services relatively
10 rapidly (Lüke and Hack, 2018).

**5 Conclusions**

The orographic complexity and lack of available high-quality hydrometric data complicated the calibration of the Z parameter and, subsequently, the validation of the model. The large variations in the Z parameter across the hydrographic basins can mainly be attributed to the forcing of Z to minimize errors between simulated and observed water yield. This forcing was
15 carried out because of the possible underestimation of precipitation. It appears that precipitation errors in the simulation of water yield were not absorbed by the Z parameter.

[revised manuscript text omitted]

Bendix, J., Rollenbeck, R., Fabian, P., Emck, P., Richter, M. and Beck, E.: Climate Variability, in Gradients in a Tropical Mountain Ecosystem of Ecuador, pp. 281–290. [online] Available from: http://www.demorgen.be/dm/nl/2461/Opinie/article/detail/1520597/2012/10/20/Jobdestructie-is-part-of-the-job.dhtml, 2008.

Breuer, L., Exbrayat, J.-F., Plesca, I., Buytaert, W., Ehmann, T., Peters, T., Timbe, E., Trachte, K. and Windhorst, D.: Global

Climate Change Impacts on Local Climate and Hydrology, in Ecosystem Services, Biodiversity and Environmental Change in a Tropical Mountain Ecosystem of South Ecuador, vol. 221, edited by J. Bendix, E. Beck, A. Bräuning, F. Makeschin, R. Mosandl, S. Scheu, and W. Wilcke, Springer Berlin Heidelberg, Berlin, Heidelberg., 2013.

Buytaert, W. and Bievre, B. De: Water for cities: The impact of climate change and demographic growth in the tropical Andes, Water Resour. Res., 48(8), 1–13, doi:10.1029/2011WR011755, 2012.

Buytaert, W., Wyseure, G., De Bièvre, B. and Deckers, J.: The effect of land-use changes on the hydrological behaviour of Histic Andosols in south Ecuador, Hydrol. Process., 19(20), 3985–3997, doi:10.1002/hyp.5867, 2005.

Buytaert, W., Iñiguez, V., Celleri, R., De Bièvre, B., Wyseure, G. and Deckers, J.: Analysis of the water balance of small páramo catchments in South Ecuador, in Environmental Role of Wetlands in Headwaters, edited by J. Krecek and M. Haigh, pp. 271–281, Springer Netherlands, Dordrecht., 2006a.

Buytaert, W., Célleri, R., De Bièvre, B., Cisneros, F., Wyseure, G., Deckers, J. and Hofstede, R.: Human impact on the hydrology of the Andean páramos, Earth-Science Rev., 79(1–2), 53–72, doi:10.1016/j.earscirev.2006.06.002, 2006b.

Buytaert, W., Celleri, R., Willems, P., Bièvre, B. De and Wyseure, G.: Spatial and temporal rainfall variability in mountainous areas: A case study from the south Ecuadorian Andes, J. Hydrol., 329(3–4), 413–421, doi:10.1016/j.jhydrol.2006.02.031, 2006c.

Buytaert, W., Iñiguez, V. and Bièvre, B. De: The effects of afforestation and cultivation on water yield in the Andean páramo, For. Ecol. Manage., 251(1–2), 22–30, doi:10.1016/j.foreco.2007.06.035, 2007.

Buytaert, W., Ce, R. and Timbe, L.: Predicting climate change impacts on water resources in the tropical Andes : Effects of GCM uncertainty, , 36, 1–5, doi:10.1029/2008GL037048, 2009.

Buytaert, W., Vuille, M., Dewulf, A., Urrutia, R., Karmalkar, A. and Célleri, R.: Uncertainties in climate change projections and regional downscaling in the tropical Andes: Implications for water resources management, Hydrol. Earth Syst. Sci., 14(7), 1247–1258, doi:10.5194/hess-14-1247-2010, 2010.

Campozano, L., Célleri, R., Trachte, K., Bendix, J. and Samaniego, E.: Rainfall and Cloud Dynamics in the Andes: A Southern Ecuador Case Study, Adv. Meteorol., 2016, doi:10.1155/2016/3192765, 2016.

Celleri, R., Willems, P., Buytaert, W. and Feyen, J.: Space–time rainfall variability in the Paute basin, Ecuadorian Andes, Hydrol. Process., 21(24), 3316–3327, doi:10.1002/hyp.6575, 2007.

Célleri, R. and Feyen, J.: The Hydrology of Tropical Andean Ecosystems: Importance, Knowledge Status, and Perspectives, Mt. Res. Dev., 29(4), 350–355, doi:10.1659/mrd.00007, 2009.

[revised manuscript text omitted]

Luna-Romero, A., Ramírez, I., Sánchez, C., Conde, J., Agurto, L. and Villaseñor, D.: Spatio-temporal distribution of

10   precipitation in the Jubones river basin, Ecuador: 1975-2013, Sci. Agropecu., 9(1), 63–70, doi:10.17268/sci.agropecu.2018.01.07, 2018.

Maldonado A., N.: La cuenca hidrográfica como espacio para promover el desarrollo sustentable: Elementos de Agroquimatología y Ecología, Loja, Ecuador., 2001.

Ministerio del Ambiente (MAE) and Ministerio de Agricultura, Ganadería, A. y P. (MAGAP): Protocolo metodológico para

15   la elaboración del Mapa de cobertura y uso de la tierra del Ecuador continental 2013 – 2014, escala 1:100.000, , 1–49 [online] Available from: http://app.sni.gob.ec/sni-link/sni/Portal SNI 2014/USO DE LA TIERRA/01-METODOLOGIA_MAPA_COBERTURA_USO.pdf, 2015.

Mora, D. E., Campozano, L., Cisneros, F., Wyseure, G., Willems, P., Divison, H., Cuenca, D., Brussel, V. U., Engineering, H., Leuven, K. U. and Divison, W. M.: Climate changes of hydrometeorological and hydrological extremes in the Paute basin

20   , Ecuadorean Andes, , 631–648, doi:10.5194/hess-18-631-2014, 2014.

Moya, R.: Climas del Ecuador, , 1–14 [online] Available from: http://186.42.174.231/gisweb/METEOROLOGIA/CLIMATOLOGIA/Climas del Ecuador 2006.pdf, 2006.

Ochoa-Tocachi, B. F., Buytaert, W. and De Bièvre, B.: Regionalization of land-use impacts on streamflow using a network of paired catchments, Water Resour. Res., 52(9), 6710–6729, doi:10.1002/2016WR018596, 2016.

25   Ochoa-Tocachi, B. F., Buytaert, W., Antiporta, J., Acosta, L., Bardales, J. D., Célleri, R., Crespo, P., Fuentes, P., Gil-Ríos, J., Guallpa, M., Llerena, C., Olaya, D., Pardo, P., Rojas, G., Villacís, M., Villazón, M., Viñas, P. and De Bièvre, B.: Data Descriptor: High-resolution hydrometeorological data from a network of headwater catchments in the tropical Andes, Sci. Data, 5, 1–16, doi:10.1038/sdata.2018.80, 2018.

Oñate, F. and Aguilar, G.: Aplicación del modelo SWAT para la estimación de caudales y sedimentos en la cuenca alta del río

30   Catamayo, in III Congreso Latinoamericano de Manejo de Cuencas Hidrográficas, p. 11, Arequipa, Perú., 2003.

Pessacg, N., Flaherty, S., Brandizi, L., Solman, S. and Pascual, M.: Getting water right : A case study in water yield modelling based on precipitation data, Sci. Total Environ., 537(December 2017), 225–234, doi:10.1016/j.scitotenv.2015.07.148, 2015.

Pourrut, P., Gómez, G., Bermeo, A. and Segovia, A.: Factores condicionantes de los regímenes climáticos e hidrológicos, in El agua en el Ecuador: Clima, precipitaciones, escorrentía, edited by P. Pourrut, O. Róvere, I. Romo, and H. Villacrés, pp. 7–

12, Quito, Ecuador., 1995.

Quintero, M., Ed.: Servicios ambientales hidrológicos en la región andina. Estado del conocimiento, la acción y la política para asegurar su provisión mediante esquemas de pago por servicios ambientales, Lima, Perú. [online] Available from: http://www.rimisp.org/wp-content/files_mf/13599885926ServiciosambientaleshidrológicosenlaRegiónAndina.pdf, 2010.

5   Redhead, J. W., Stratford, C., Sharps, K., Jones, L., Ziv, G., Clarke, D., Oliver, T. H. and Bullock, J. M.: Empirical validation of the InVEST water yield ecosystem service model at a national scale, Sci. Total Environ., 569–570, 1418–1426, doi:10.1016/j.scitotenv.2016.06.227, 2016.

Richter, M. and Moreira, A.: Heterogeneidad climática y diversidad de la vegetación en el sur de Ecuador : un método de fitoindicación, Biologia (Bratisl)., 12(2), 217–238 [online] Available from:
10   http://www.scielo.org.pe/pdf/rpb/v12n2/v12n2a07.pdf, 2005.

Rollenbeck, R. and Bendix, J.: Rainfall distribution in the Andes of southern Ecuador derived from blending weather radar data and meteorological field observations, Atmos. Res., 99(2), 277–289, doi:10.1016/j.atmosres.2010.10.018, 2011.

Sánchez-Canales, M., López Benito, A., Passuello, A., Terrado, M., Ziv, G., Acuña, V., Schuhmacher, M. and Elorza, F. J.: Sensitivity analysis of ecosystem service valuation in a Mediterranean watershed, Sci. Total Environ., 440, 140–153,
15   doi:10.1016/j.scitotenv.2012.07.071, 2012.

Sharp, R., Tallis, H., Ricketts, T., Guerry, A., Wood, S., Chaplin-Kramer, R., Nelson, E., Ennaanay, D., Wolny, S., Olwero, N., Vigerstol, K., Pennington, D., Mendoza, G., Aukema, J., Foster, J., Forrest, J., Cameron, D., Arkema, K., Lonsdorf, E., Kennedy, C., Verutes, G., Kim, C. and Gua, J.: InVEST +VERSION+ User's Guide, , 334 [online] Available from: http://data.naturalcapitalproject.org/nightly-build/invest-users-guide/html/, 2018.

20   Tallis, H. and Polasky, S.: Mapping and valuing ecosystem services as an approach for conservation and natural-resource management, Ann. N. Y. Acad. Sci., 1162, 265–283, doi:10.1111/j.1749-6632.2009.04152.x, 2009.

Vázquez, R. F.: Modelación hidrológica de una microcuenca Altoandina ubicada en el Austro Ecuatoriano, Maskana, 1(1), 79–90 [online] Available from: https://publicaciones.ucuenca.edu.ec/ojs/index.php/maskana/article/view/370/315, 2015.

Vigerstol, K. L. and Aukema, J. E.: A comparison of tools for modeling freshwater ecosystem services, J. Environ. Manage.,
25   92(10), 2403–2409, doi:10.1016/j.jenvman.2011.06.040, 2011.

World Meteorological Organization (WMO): Guide to Climatological Practices WMO-No. 100., 2011.

Xu, X., Liu, W., Scanlon, B. R., Zhang, L. and Pan, M.: Local and global factors controlling water-energy balances within the Budyko framework, Geophys. Res. Lett., 40(December), 6123–6129, doi:10.1002/2013GL058324, 2013.

Zhang, C., Li, W., Zhang, B. and Liu, M.: Water yield of Xitiaoxi River basin based on InVEST modeling, J. Resour. Ecol.,
30   3(2008), 50–54, doi:10.5814/j.issn.1674-764x.2012.01.008, 2012.

Zhang, L., Hickel, K., Dawes, W. R., Chiew, F. H. S., Western, A. W. and Briggs, P. R.: A rational function approach for estimating mean annual evapotranspiration, Water Resour. Res., 40(2), W025021–W02502114, doi:10.1029/2003WR002710, 2004.

---

## Author Comment (AC2) · 26 Dec 2018

Comment: The manuscript by Minga-León and coauthors provides an estimation of water production in nine hydrographic basins in southern Ecuador. Without offering a complete review of the article, which would be given during the peer-review process, I was surprised to see that the literature review, especially the one offered in the Introduction, presents references that date back to 2015 at the latest. Apart from Redhead et al (2016) and Li et al. (2018), both referring to the InVEST model, there are not updated citations on tropical, Andean, and Amazonian hydrology that can complement, contextualise, and offer further discussion to enrich this work. I provide here six relevant articles that the authors could read, include, and use as a starting point to strengthen their literature review and scientific content for their study.

[Figure]

Response: We appreciate the literature recommendations. After reviewing the proposed references, we considered that it was appropriate to include two references relevant for our study to strengthen and enrich our understanding of the physical context and scientific studies in the region. The following works were referenced in the introduced (Pg. 2):

- Ochoa-Tocachi, B. F., Buytaert, W., De Bièvre, B., Célleri, R., Crespo, P., Villacís, M., Llerena, C. A., Acosta, L., Villazón, M., Guallpa, M., Gil-Ríos, J., Fuentes, P., Olaya, D., Viñas, P., Rojas, G., and Arias, S. (2016) Impacts of land use on the hydrological response of tropical Andean catchments. Hydrol. Process., 30: 4074–4089. doi: 10.1002/hyp.10980.

- Ochoa-Tocachi, B. F., Buytaert, W., Antiporta, J., Acosta, L., Bardales, J. D., Célleri, R., Crespo, P., Fuentes, P., Gil-Ríos, J., Guallpa, M., Llerena, C., Olaya, D., Pardo, P., Rojas, G., Villacís, M., Villazón, M., Viñas, P., and De Bièvre, B. (2018) High-resolution hydrometeorological data from a network of headwater catchments in the tropical Andes. Sci. Data, 5: 180080. doi: 10.1038/sdata.2018.80

---

## Referee Comment (RC2) · Anonymous Referee #2 · 18 Feb 2019

This study examines the use of the Budyko framework based InVEST model to estimate water yield of the hydrographic basins located in the southern region of Ecuador. Based on hydro-climatic observations (precipitation, PET and runoff) and geo-physical attributes (soil/vegetation depth), the authors established the model by calibrating the single eco-hydrologic parameter (Z) of the InVEST model. Across the study area, the authors then identified basins with highest (and lowest) water production. While I welcome the author's contribution on predicting water yield in the complex terrain and data-scarce regions, at the same time I see number of issues with the current manuscript (mentioned below).

My main concern with this work is on the author's use of the Budyko framework for estimating water yield based on the functional form of one parameter Budyko model

(Fu's equation; $\omega$) without any proper validation. A recent study by Padron et al., 2017 (https://doi.org/10.1002/ 2017WR021215) provides a comprehensive picture on control of $\omega$ – relationships of which to catchment geo-physical attributes are not very clear (i.e., they appears to be location/climate specific). Therefore before resorting to any sort of the functional relationship (for $\omega$), it needs to be properly validated. The authors must show some sort of validations through e.g., split sampling test in time and space. Besides, it is not clear to me why the authors do not directly estimate the $\omega$ values through calibration. Such procedure is very common in literature (see the references given in Padron et al. 2017; https://doi.org/10.1002/2015GL066363; https://doi.org/10.1002/2015GL066363). I would like to see more discussion on this topic and especially the rational of author's selection (for the Budyko form).

Another major concern, I have with this study is the achieved overall modeling results. Considering even the functional relationship of $\omega$ (and Z to estimate) based on the outflows of 9 basins, results shown in Table 4 rather indicate very poor model fits in 4 basins; and other 2 have unreasonably low Z values (less than 5) and one at the border line of Z = 5. The authors then left with 2 basins in which the Z parameter can be reliably estimated; and based on this I do not see how you come up with the conclusion that "The modeling of water yield in the majority of hydrographic basins was satisfactory". Besides there is no information provided in the manuscript on how the Z parameters estimated in limited number of (sub-) basins are applied to the entire (hydrographic) region – or even at the pixel level (Figures 5 & 6)? How did you treat the bad preforming basins (in terms of it and unreasonable Z values)?

Page 7: It is not entirely true that "Data on the root restriction layer were unavailable" as authors, stated. Specifically in the HWSD database, which the authors are using – there is information on the root restrictions in so-called, attribute "ROOTS" (http://www.fao.org/docrep/018/aq361e/aq361e.pdf). Please double check. Also I think there is some mismatch between the authors plotted soil-depth (in Figure 3) and ones given in information of the HWSD database. In the manual of the HWSD, the

"REF_DEPTH" attribute is defined as: "Reference soil depth of all soil units are set at 100 cm, except for Rendzinas and Rankers of FAO-74 and Leptosols of FAO-90, where the reference soil depth is set at 30 cm, and for Lithosols of FAO-74 and Lithic Leptosols of FAO-90, where it is set at 10 cm. An approximation of actual soil depth can be derived through accounting for relevant depth limiting soil phases, obstacles to roots and occurrence of impermeable layers (the latter two refer to ESDB only)". Besides it is not clear that how the authors use the information of the soil depth (from HWSD) and the Root depth (in Table 1) in estimating the Z parameter (or in AWC). Please clarify these points.

Table 3: Are the (irrigation) flow estimates being constant over the study period?

---

## Author Comment (AC3) · 13 Mar 2019

Dear reviewer: We are thankful for the time taken to review our manuscript, and we consider that the questions and comments are appropriate.

1) Comment: My main concern with this work is on the author's use of the Budyko framework for estimating water yield based on the functional form of one parameter Budyko model (Fu's equation; $\omega$) without any proper validation. A recent study by Padron et al., 2017 (https://doi.org/10.1002/ 2017WR021215) provides a comprehensive picture on control of $\omega$ – relationships of which to catchment geo-physical attributes are not very clear (i.e., they appears to be location/climate specific). Therefore before resorting to any sort of the functional relationship (for $\omega$), it needs to be properly vali-

dated. The authors must show some sort of validations through e.g., split sampling test in time and space. Besides, it is not clear to me why the authors do not directly estimate the $\omega$ values through calibration. Such procedure is very common in literature (see the references given in Padron et al. 2017; https://doi.org/10.1002/2015GL066363; https://doi.org/10.1002/2015GL066363). I would like to see more discussion on this topic and especially the rational of author's selection (for the Budyko form).

Response: Yes, we agree with you. Calibration and validation must be done, and in fact, have been done. We will explain bellow. The present study used the Water Yield model from InVEST, for the estimation of water yield, which is based on the hydrological framework of Budyko adapted by Fu (1981) and Zhang et al. (2004). The first incorporates a catchment parameter (w) and the second an empirical parameter (Z). It's important to mention that the Water Yield model of InVest is designed to model long term averages. As a rule of thumb, a 10-year period should be used to capture some climate variability, according to Sharp et al. (2018).Due to the lack of continuity and sample size of observed runoff of some hydrometrics stations, the available data allowed only making calibration and validation of the model at the following gauges; Alamor (H0616), Puyango (H0591), Arenillas (H0574), Jubones (H0530) and Zamora (H0889), in Table 1. Regarding hydrometric stations that have few observed data such as Macará (H0626), Catamayo (HB32), Chaguana (H0508) and Mayo (H0966) only calibration was performed. To standardize and summarize the information, it was decided to show only the global results over the whole period of available data.

Why not estimate directly the w factor?

The InVEST user's guide (Sharp 2018), between one of the prosed methods is to calibrate the parameter Z of the model which varies from 1 to 30 which will allow simultaneously to calculate w (equation 4). This method was evaluated and recommended by Hamel and Guswa (2015) and has been applied in studies such as Pessacg et al. (2015) and Redhead et al. (2016).

2) Comment: Another major concern, I have with this study is the achieved overall modeling results. Considering even the functional relationship of $\omega$ (and Z to estimate) based on the outflows of 9 basins, results shown in Table 4 rather indicate very poor model fits in 4 basins; and other 2 have unreasonably low Z values (less than 5) and one at the border line of Z = 5. The authors then left with 2 basins in which the Z parameter can be reliably estimated; and based on this I do not see how you come up with the conclusion that "The modeling of water yield in the majority of hydrographic basins was satisfactory". Besides there is no information provided in the manuscript on how the Z parameters estimated in limited number of (sub-) basins are applied to the entire (hydrographic) region – or even at the pixel level (Figures 5 & 6)? How did you treat the bad preforming basins (in terms of it and unreasonable Z values)?.

Response: Indeed, in the conclusions we stated that the results are satisfactory in most basins. This statement is supported (without mentioning it in the manuscript) by the classification of Moriasi et. al (2007). Indeed, according to these authors, the simulation results based on the relative error (PBIAS) are very good when PBIAS is $<\pm 10\%$, good between 10 and $\pm 15\%$, satisfactory between $\pm 15$ and $\pm 25\%$ and unsatisfactory $>\pm 25\%$. Water Yield Model is a relatively simple model (with one parameter). The unsatisfactory results in some basins can be explained only for two reasons, 1) whether the model does not adequately describe the studied phenomenon and/or 2) that the input data are not enough to describe the phenomenon or of poor quality (non-optimal meteorological network, no consideration of change of land use, observed flows of doubtful quality).

3) Comment: Page 7: It is not entirely true that "Data on the root restriction layer were unavailable" as authors, stated. Specifically in the HWSD database, which the authors are using – there is information on the root restrictions in so-called, attribute "ROOTS" (http://www.fao.org/docrep/018/aq361e/aq361e.pdf). Please double check. Also I think there is some mismatch between the authors plotted soil-depth (in Figure 3) and ones given in information of the HWSD database. In the manual of the HWSD,

the REF_DEPTH attribute is defined as: Reference soil depth of all soil units are set at 100 cm, except for Rendzinas and Rankers of FAO-74 and Leptosols of FAO-90, where the reference soil depth is set at 30 cm, and for Lithosols of FAO-74 and Lithic Leptosols of FAO-90, where it is set at 10 cm. An approximation of actual soil depth can be derived through accounting for relevant depth limiting soil phases, obstacles to roots and occurrence of impermeable layers (the latter two refer to ESDB only). Besides it is not clear that how the authors use the information of the soil depth (from HWSD) and the Root depth (in Table 1) in estimating the Z parameter (or in AWC). Please clarify these points.

Response: We agree. The HWSD database manage an attribute named "Obstacles to Roots", however this data are not available to most of the countries and specifically to the study area (Figure 1). That is the reason why we decided to use the soil depth as recommended by Sharp et al. (2018) as an approximation to the depth of restriction of the roots. The data of depth of soil and depth of rooting of the vegetation are used in the estimation of PAWC, variable requested by the Water Yield model. AWC values (mm) were obtained from the HWSD database, and these values were divided by the minimum value of the root restriction depth or rooting depth of vegetation (mm) with the goal of obtaining the required fraction (PAWC) by the model. According to Sharp et al. (2018) "the model determines the minimum of root restricting layer depth and rooting depth for an accessible soil profile for water storage". The PAWC values are dimensionless (0 to 1) and are basically obtained by solving equation 5 in the document. PAWC=AWC/(Min (Rest.layer.depth,root.depth))

4) Comment: Table 3: Are the (irrigation) flow estimates being constant over the study period?

Response: Local water transfers (irrigation) were considered only in the Catamayo and Macará basins in the calibration process. The calibration depended on the availability of data for each hydrometric station.

New reference

Moriasi, D., Arnold, J. and Liew, M. W. Van: Model Evaluation Guidelines for Systematic Quantification of Accuracy in Watershed Simulations, Trans. ASABE, 50(3), 885–900, doi:10.13031/2013.23153, 2007.

[Figure]

[Figure]

**Fig. 1.** The HWSD database

Table 1. Calibration and validation

| Station code | Hydrometric station | Area (Km²) | Process | Calibration period | Excluded years | Observed water yield | Estimated water yield | Z | Error relativo |
|---|---|---|---|---|---|---|---|---|---|
| H0616 | Alamor at Saucillo (Dj Celica) | 585 | Calibration | 1970-1999 | | 405 | 407 | 13 | 0.58 |
| | | | Validation | 2000-2011 | | 357 | 325 | 13 | -8.90 |
| H0591 | Puyango at Cpto. Militar (Pte. Carretera) | 2728 | Calibration | 1970-1999 | | 1017 | 769 | 1 | -24.31 |
| | | | Validation | 2000-2011 | | 1094 | 832 | 1 | -23.96 |
| H0574 | Arenillas at Arenillas | 493 | Validation | 1970-1983 | 1984-1990 | 538 | 570 | 3 | 5.87 |
| | | | Calibration | 1991-2011 | 1984-1990 | 415 | 423 | 3 | 1.85 |
| H0530 | Jubones at Ushcurrumi | 3636 | Validation | 1970-1980 | | 472 | 377 | 4 | -20.11 |
| | | | Calibration | 1981-2011 | | 406 | 438 | 4 | 8.01 |
| H0889 | Zamora DJ Sabanilla (at Zamora) | 1422 | Calibration | 1982-2000 | 1985 | 1660 | 828 | 1 | -50.09 |
| | | | Validation | 2001-2011 | 2002 | 1841 | 1156 | 1 | -37.17 |

**Fig. 2.** Table 1. Calibration and validation